# The Footprint of Type 1 Diabetes on Red Blood Cells: A Metabolomic and Lipidomic Study

**DOI:** 10.3390/jcm12020556

**Published:** 2023-01-10

**Authors:** José Raul Herance, Andreea Ciudin, Rubén Lamas-Domingo, Carolina Aparicio-Gómez, Cristina Hernández, Rafael Simó, Martina Palomino-Schätzlein

**Affiliations:** 1Medical Molecular Imaging Research Group, Vall d’Hebron Research Institute and Autonomous University of Barcelona, 08035 Barcelona, Spain; 2CIBER-bbn (ISCIII), 28040 Madrid, Spain; 3Diabetes and Metabolism Research Unit, Vall d’Hebron Research Institute, Autonomous University of Barcelona, 08035 Barcelona, Spain; 4CIBERDEM (ISCIII), 28040 Madrid, Spain; 5NMR Facility, Centro de Investigación Príncipe Felipe, 46013 Valencia, Spain; 6ProtoQSAR SL, CEEI (Centro Europeo de Empresas Innovadoras), Parque Tecnológico de Valencia, 46980 Valencia, Spain

**Keywords:** type 1 diabetes, NMR spectroscopy, metabolomics, lipidomics, red blood cells

## Abstract

The prevalence of diabetes type 1 (T1D) in the world populations is continuously growing. Although treatment methods are improving, the diagnostic is still symptom-based and sometimes far after onset of the disease. In this context, the aim of the study was the search of new biomarkers of the disease in red blood cells (RBCs), until now unexplored. The metabolomic and the lipidomic profile of RBCs from T1D patients and matched healthy controls was determined by NMR spectroscopy, and different multivariate discrimination models were built to select the metabolites and lipids that change most significantly. Relevant metabolites were further confirmed by univariate statistical analysis. Robust separation in the metabolomic and lipidomic profiles of RBCs from patients and controls was confirmed by orthogonal projection on latent structure discriminant analysis (OPLS-DA), random forest analysis, and significance analysis of metabolites (SAM). The main changes were detected in the levels of amino acids, organic acids, creatine and phosphocreatine, lipid change length, and choline derivatives, demonstrating changes in glycolysis, BCAA metabolism, and phospholipid metabolism. Our study proves that robust differences exist in the metabolic and lipidomic profile of RBCs from T1D patients, in comparison with matched healthy individuals. Some changes were similar to alterations found already in RBCs of T2D patients, but others seemed to be specific for type 1 diabetes. Thus, many of the metabolic differences found could be biomarker candidates for an earlier diagnosis or monitoring of patients with T1D.

## 1. Introduction

Diabetes mellitus is the first non-communicable disease that the United Nations recognized as a 21st-century pandemic problem [1]. Diabetes is a group of diseases mainly characterized by high blood glucose levels. Type 1 diabetes (T1D) and type 2 diabetes (T2D) both involve hyperglycemia, but are disorders with different patterns, mechanisms, clinical presentations, and disease progressions. Nevertheless, once hyperglycemia occurs, patients with all forms of diabetes can develop similar chronic complications, in different rates of progression and form [2].

T2D accounts for about 80–90% of the cases of diabetes and is due to a progressive loss of β-cell mass and insulin secretion, on the background of insulin resistance (IR), low-grade inflammation, and oxidative stress [3]. In 2019, 463 million adults (20–79 years) were living with diabetes worldwide and this number could rise to 700 million by 2045 “www.idf.org (accessed on 6 January 2023)”. Due to the latent onset of this disease, there is also a long pre-detection period, and up to one-half of cases in the population may be undiagnosed [4]. 

T1D accounts for about 5–10% of all cases of diabetes and is due to the autoimmune destruction of beta cells of the pancreas, leading to a total incapacity of producing insulin [5,6]. The incidence and prevalence of T1D are increasing worldwide dramatically. While T2D starts generally at an older age (average diagnosis age 45–52 years) [7], T1D tends to affect much younger people (average diagnosis age 10–14 years) [8], children in many cases, although the age limit is not mandatory. Patients with T1D need complete substitutive treatment with insulin for their lifetime (both long-acting and rapid-acting insulin) and have been classically described to have unimpaired insulin resistance (Imran 2008). Nevertheless, recent scientific reports suggest that people with T1D also develop insulin resistance (IR), which is related to complications and comorbidities in patients [9]. 

Furthermore, mechanisms such as lipotoxicity, glucotoxicity, closely related to hyperglycemia [10], and iatrogenic hyperinsulinemia [11] cause oxidative stress that can contribute to the development of IR in T1D. In oxidative stress, reactive oxygen species (ROS) can induce insulin receptor substrate serine/threonine phosphorylation, impair cellular redistribution of insulin signaling components, reduce *GLUT4* gene transcription, or alter mitochondrial activity [12]. On the other hand, oxidative stress can increase the production of pro-inflammatory cytokines and induce inflammatory processes. Immune cells such as macrophages, which are the source of pro-inflammatory cytokines, can migrate to pancreatic islet cells [13]. Thus, chronic inflammation can also contribute to the development of IR in T1D. For instance, adipose cell enlargement can lead to a pro-inflammatory state and the formation of pro-inflammatory compounds, such as interleukin-6, C-reactive protein, and plasminogen activator inhibitor-1, which, in turn, contribute to the death of beta cells, modulation of beta cell regeneration processes, and IR [14,15]. 

Even if the main causes of IR in T1D have been explored, at present, there is a lack of data that connect the risk factors of IR in T1D with the metabolomic and lipidomic profile. Glycated hemoglobin (HbA1c) is the parameter used worldwide for the diagnosis and follow-up of the metabolic control of diabetes [16]. HbA1c is adult hemoglobin (HbA) with glucose bound to its b chain N-terminal valine, resulting from non-enzymatic glycation in erythrocytes, also known as red blood cells (RBCs) [17].

RBCs are the most abundant cell type in the human body and play essential roles in metabolism, transport, and respiration. Their composition, shape, abundance, and motility can be altered by diseases. Several studies revealed that RBCs suffer general alterations in diabetes [18]. For instance, RBCs’ aggregability increase [19] as well as alterations of RBC shape [20], deformability [21], and membrane phospholipid composition [22]. In addition, the glutathione redox system is affected [23] with decreases in reduced glutathione (GSH), GSH peroxidase and reductase activity, energetic metabolism, as well as alterations in adenosine triphosphate (ATP) levels [24].

Our group recently showed that the metabolomics profile of RBCs is generally altered in T2D disease [25], mainly affecting the redox and energetic metabolism, as well as the transport of certain metabolites necessary for physiological functions. However, at present, no information about the impact of T1D on the metabolomics profile of RBCs is known. On these bases, the aim of this study was to explore the general metabolic alteration of RBCs in patients with T1D to find fingerprints of insulin resistance and metabolic information for monitoring this type of patients.

## 2. Materials and Methods

### 2.1. Study Design and Subjects

The study was designed as a cross-sectional study case-control. It was approved by the local Ethics Committee (PR(AG)234/2015) and carried out in accordance with the Declaration of Helsinki. Patients were recruited from the outpatient clinic of the Endocrinology Department of Vall d’Hebron University Hospital and divided into two groups as follows: 20 patients with T1D matched by age, sex, and body mass index (BMI) with 20 healthy individuals. All patients signed the informed consent form before their inclusion in the study.

The inclusion criteria were the following: (a) age > 18 years, (b) T1D, (c) written informed consent form. The exclusion criteria were the following: (a) any other type of diabetes, except for T1D (T2D, MODY, etc.), (b) tobacco use in the last 12 months, (c) alcohol intake of more than 30 g in males and 20 g in females, (d) hemoglobinopathies, (e) hematologic disease, (f) any other disease or treatment that might influence the RBCs’ metabolism (chronic kidney disease, cardiac valve prosthesis, chemotherapy, anticoagulants, etc.). All diabetic patients had positive islet autoantibodies of anti-GAD-65 (>50 IU/mL).

This was a pilot study. At present, there are no available data regarding the impact of T1D on the RBCs’ metabolism; therefore, we were not able to perform a sample size calculation. Thus, the size has been selected by taking into account ethical considerations and the pilot studies in T2D [25,26]. 

All the subjects included in the study underwent a complete medical history review, anthropometric evaluation, and biochemical analysis. Anthropometrical parameters of weight (kg), waist (cm), and height (m) were measured just before the collecting of the blood samples. Biochemical analysis was performed using standardized and validated routine methodologies on the blood samples collected at fasting conditions at the biochemistry core facilities of Vall d’Hebron University Hospital. Plasmatic concentrations of glucose (mg/dL), glycosylated hemoglobin (HbA1c) (%DCCT), high-density lipoprotein (HDL) (mg/dL), low-density lipoprotein (LDL) (mg/dL), triglycerides (TG) (mg/dL), anti-GAD65 (IU/mL), and insulin (mU/mL) were measured in all the subjects. Body mass index (BMI) kg/m^2^ was calculated using the following formula: body weight (kg)/height^2^ (m) [27]. 

### 2.2. Peripheral Blood Samples

Blood samples were collected followed a similar protocol as in previous studies [25,26]. Briefly, peripheral blood was collected under fasting conditions, stored at 4 °C, and processed within the first hour. Then, 5 mL of peripheral blood were poured into a quartz tube with 10 mL of Ficoll until 2 phases were separated by gravity, around 30 min. Then, the pellet was washed twice with 10 mL of cool Hank’s Balanced Salt Solution in a centrifuge at 200× *g* and 4 °C for 10 min without breaks. Cell counting was performed on a Neubauer chamber and the purity was tested with flow cytometry. For storage, 200 µL of ice-cold methanol were added per 10 million cells for quenching and the samples were frozen directly at −80 °C. 

### 2.3. Extraction of Metabolites and Lipids

Metabolites were extracted from RBCs as described previously [25]. Briefly, frozen samples were allowed to thaw for 5 min on ice. Then, 800 µL of chloroform at 4 °C were added to each sample tube. After 10 min, the samples were homogenized, resuspended with a pipette, and transferred to a bigger plastic tube. Then, samples were placed in liquid nitrogen for 1 min and then allowed to thaw on ice for 2 min, for uniform cell membrane breakage. This step was performed two more times. Afterwards, 1250 µL of distilled water and 1250 µL of chloroform were added and the sample vortexed. Samples were then centrifuged at 13,000× *g* for 20 min at 4 °C to separate the phases. The upper phase contained polar metabolites in a mixture of water/methanol and was lyophilized overnight. The lower chloroform phase containing the lipids was evaporated under nitrogen flux. Extracts were stored at −80 °C until NMR sample preparation and analysis. For NMR sample preparation, the frozen samples were allowed to thaw for 5 min on ice. Then, 550 µL of NMR buffer (100 mM Na_2_HPO_4_ in D_2_O at pH 7.4) containing 0.1 mM deuterated trimethylsilylpropanoic acid as an internal standard were added to the polar samples and transferred to a 5 mm NMR tube. For the preparation of the lipidomics samples, 600 µL of deuterated chloroform containing 0.003% *v*/*v* tetramethylsilane was added to the lipidic samples and transferred to a 5 mm NMR tube. All samples were analyzed the same day and stored at 4 °C until the analysis.

### 2.4. NMR Experiments

NMR spectra were recorded as described previously [25] at 27 °C on a Bruker AVII-600 spectrometer using a 5 mm triple resonance cryoprobe and processed using TopSpin version 3.2 software (Bruker BioSpin). Before starting sample measurement, the temperature was calibrated at 27 °C with a 99.8% MeOD sample. Further, the parameters of the spectrometer were optimized to ensure an optimal resolution and sensitivity (width at half-height ≤ 1 Hz), with a standard sample containing 2 mM of sucrose, 0.5 mM of DSS (sodium trimethylsilylpropanesulfonate), 2 mM of NaN_3_ in 90% H_2_O, and 10% D2O. ^1^H 1D NOESY NMR spectra were acquired with 256 free induction decays (FIDs). A 4s relaxation delay was incorporated between FIDs and the water signal was eliminated with presaturation in the aqueous samples. The FID values were multiplied by an exponential function with a 0.5 Hz line broadening factor for an optimal baseline correction, with 64K data points digitalized over a spectral width of 30 ppm. Quality control samples, consisting of a mixture of lactate, creatine, citric acid, phenylalanine, and glucose, were run in regular intervals to confirm the reproducibility of the measurements. 

### 2.5. Data Analysis

^1^H-NMR signals from the spectra were assigned with the help of previous data [25,28] and the spectral databases Human Metabolome Database and Biological Magnetic Resonance Bank [29,30]. In ambiguous cases, the assignment was confirmed by spiking the spectra with reference compounds. For metabolite quantification, spectra were normalized to total intensity. This allowed us to work with normalized concentration values and avoid the detection of changes related exclusively to differences in total sample amount and experimental error during the extraction process. The predominant glucose signals and the solvent signals were excluded from normalization. Optimal integration regions were defined for each metabolite, an attempt being made to select the signals without overlapping (Appendix A). Integration was performed with global spectrum deconvolution in MestreNova 12 (Mestrelab Research S. L.). The datasets are available in the Zenodo repository [31].

For modelling, metabolite integration data were Pareto-scaled (each value being divided by the square root of the standard deviation of each variable) and mean-centered, for an easier interpretation of the data and to also take into account the variation of small signals. Orthogonal projection on latent structure discriminant analysis (OPLS-DA) and projection on latent structure (PLS) regression were performed with SIMCA-P 14.0 (Umetrics, Sweden). OPLS-DA and PLS models were described with R2Y(cum) (representing the cumulative SS of all the y-variables explained by the extracted components) and Q2(cum) (describing the cumulative Q2 for all y-variables for the extracted components). Score plots (representing scores related to prediction Y t(1) versus scores related to the first orthogonal component t0) and S-plots (plotting the modelled covariation (p(1)) versus the modelled correlation (p(corr)) [32] were used to detect significant metabolites and lipids in the OPLS-DA models. The glucose signal, due to its high intensity and correlation to plasmatic glucose, was excluded from OPLS-DA analysis. OPLS-DA and PLS models were validated in SIMCA-P by a 100-times permutation (where the overfit of the model is measured by the intercept of the regression line of the correlation coefficient between the original y-variable and the permuted y-variable on the x-axis versus the cumulative R2 and Q2 on the y-axis) and analysis of variance testing of cross-validated predictive residuals (CV-ANOVA) [33]. CV-ANOVA is a significance test for the Q2YCV cross-validation using the F-distribution, based on an ANOVA assessment of the cross-validatory (CV) predictive residuals of the model. In PLS analysis, metabolites that were relevant for the correlation were identified by VIP (variable-importance-projection) values >1.

Significance analysis of metabolites (SAM) analysis was performed with Metaboanalyst [34] applying the *siggenes* package [35]. To each variable, a significance score was assigned based on its change relative to the standard deviation of repeated measurements. For a variable with scores greater than an adjustable threshold, its relative difference is compared to the distribution estimated by random permutations of the class labels. For each threshold, a certain proportion of the variables in the permutation set will be found to be significant by chance. The proportion is used to calculate the FDR. In our analysis, a Delta value of 1.4 was defined to control FDR.

Random Forest (RF) analysis was performed with Metaboanalyst [34] based on the *randonForest* package [36]. RF uses groups of classification trees, each of which is grown by random feature selection from a bootstrap sample at each branch. Class prediction is based on the majority vote of the ensemble. For initial tree construction, one-third of the samples are left out for validation. The OOB (out-of-bag) data are then used as a test sample to obtain an unbiased estimate of the classification error (OOB error). Variable importance is calculated by the measure of the increase of the OOB error when it is permuted.

Hierarchical clustering was performed with Metaboanalyst with the hclust function in package stat and the result represented as a heatmap.

Box plots and *t*-tests were performed with normalized concentration data in Graphpad Prism 8. Only metabolites that showed high scores in all three models (OPLS-DA, Random Forest, and SAM) were considered for univariate analysis.

For the biological interpretation of the results and the identification of metabolic pathways, the Kegg Data Base and MetPA (Metaboanalyst) were used.

## 3. Results

Table 1 summarizes the main characteristics of the subjects and controls included in the study.

A metabolomic and a lipidomic analysis of RBCs of all individuals was performed by proton nuclear magnetic resonance (^1^H-NMR). As a result, 68 different hydrophilic metabolites and 12 different types of lipid groups were identified and quantified (Appendix A). Three different discrimination methods were applied in order to analyze significant differences between both groups: OPLS-DA (Orthogonal Projections to Latent Structures Discriminant Analysis), Random Forest Classification, and SAM (Significance Analysis of Metabolites). Significant discrimination between healthy controls (CT) and T1D patients could be obtained for the three model types of the metabolomic and lipidomic profiles (Figure 1).

In addition, a heatmap analysis was performed to have an overview of the main metabolic changes in the patient and control groups. As can be seen in Figure 2A, a similar tendency in metabolite levels in T1D patients is detected (yellow color), while controls have a higher dispersion (blue color). As a general tendency, T1D patients seem to have lower levels of amino acids and organic acids.

Concerning lipid metabolism, T1D patients tended to have lower levels of unsaturated fatty acids (Figure 2B). This tendency was, however, less pronounced for polyunsaturated fatty acids (PUFA). Additionally, phosphatidylcholines were reduced as well as phospholipids in general. On the other hand, a decrease of lipid chain length could be detected in T1D patients, determined by the ratio of lipid methyl groups (Lipid CH3-, increase) versus linear lipid chains (Lipid–CH2, decrease).

In order to make a robust metabolite selection, metabolites that had high scores in all three model types (Figure 1) were further submitted to univariate analysis. The resulting significant metabolites are represented in Figure 3, and can be divided into different metabolite groups. As can be seen, organic acids and metabolites related to lipid and choline metabolism seem to be the most affected.

In order to detect which of the metabolic changes were directly correlated with plasma glucose levels in fasting conditions, we performed PLS regression analyses of the metabolomic and the lipidomic profile versus glucose concentrations. As a result, we obtained a good correlation of glucose with several metabolites (Appendix A) and a moderate correlation with some blood lipids (Appendix A). The metabolites and the lipids that were relevant for the correlation (VIP > 1) are listed in Appendix A. Most metabolites that were significantly altered in RBCs of T1D patients also had a correlation with plasma glucose levels. Only glycerophosphocholine (GPC), inosinic acid (IMP), and creatine were not relevantly correlated. Regarding lipidomics, although the correlation was much weaker, all relevant lipid groups (lipid CH_3_, lipid CH_2_, and phosphatidylcholine) seemed to be related to plasma glucose concentration.

In order to correlate the metabolite changes with alterations of specific metabolic pathways, an enrichment pathway analysis was performed (Figure 4), showing that energy metabolism, fatty acid metabolism, and amino acid metabolism were the most affected metabolic pathways in T1D compared to healthy controls.

## 4. Discussion

In the present pilot study, we detected a clear difference in the metabolomic and lipidomic profile of patients with T1D and control subjects, proving that T1D has a specific impact on RBC metabolism and transport. Our group has previously reported that the metabolomics profile of RBCs is generally altered in T2D disease [25], mainly affecting redox and energy metabolism, as well as the transport of certain metabolites with physiological functions. As far as we know, this is the first study that shows a significant impact of T1D on the metabolomics and lipidomics profile of RBCs.

In our analysis, we detected a significant reduction of acetate in patients, as well as a tendency of lactate to decrease (Figure 2A). This could reflect a reduction in glycolysis, a simple pathway of glucose metabolism that is regulated by insulin secretion and sensibilization [37]. Previous studies reported increased plasmatic lactate and acetate levels in patients with diabetes [38]. Thus, the increment in lactate could be related to increased carbonic anhydrase activity in RBCs, transporting a higher amount of lactate into blood plasma [39]. Our group reported similar findings in patients with T2D [25], which could indicate that similar processes happen in T1D and T2D.

Other metabolites altered in RBCs were isoleucine and leucine, essential metabolites for the production and formation of hemoglobin and the production of RBCs [40]. This alteration is similar to that detected in T2D patients [25], where the BCCAs valine and leucine decreased. An opposite tendency was detected in plasma of diabetic patients where the concentration of BCAAs was increased in both T1D and T2D [41,42]. This inverse correlation could be related to the fact that RBCs with highly glycosylated hemoglobin, as occurs in both T1D and T2D diabetes, seem to have a lower BCAA uptake capacity. Interestingly, isoleucine was exclusively altered in RBCs of T1D patients, while significant changes in valine were only detected in T2D. The differences in the alteration of BCCAs depending of the type of diabetes could indicate a different pathophysiology associated with BCAAs in both types of diabetes, since this type of amino acids is transported by RBCs. Furthermore, valine and leucine could be RBCs’ fingerprints of T2D and T1D, respectively. In addition, the alteration of the organic acids 2-hydroxyisovalerate and 3-mehtyl-2-oxovalerate, key compounds in BCAAs metabolism, were only decreased in T1D, indicating a higher affectation of BCAAs in this type of diabetes.

Previous studies reported changes in other kinds of amino acids associated with diabetes and pre-diabetes in many studies [43]. Interestingly, the changes in non-BCAA amino acids seemed to be less pronounced in RBCs of T1D patients than in T2D [25]. For instance, alanine, asparagine, glutamate, proline, threonine, and methionine seem not to be affected by T1D, while they change significantly in T2D. These altered amino acid levels may directly interfere with insulin-regulated glycogen synthesis, and the lower impact on amino acid concentration in RBCs may be related to a lower level of IR in T1D, reducing the enhancement of plasma concentration of these type of metabolites [44]. However, further studies are mandatory to clarify this assumption.

Another noteworthy result from our study was the increase of creatine and phosphocreatine in RBCs of T1D patients. Plasmatic creatine has been recently identified as a potential biomarker for mitochondria dysfunction and for T2D risk in the PREVEND study from Post et al [45]. Our results indicate that this increase could be more generalized in the organism, taking place in cells, and may be related to alterations in muscle or energy metabolism where creatine plays an important role. This is further supported by the increase of inosine monophosphate (IMP) showed in RBCs of T1D patients, a nucleotide related to energy and generated in muscles during exercise [46]. Interestingly, neither creatine nor IMP showed a direct correlation with blood glucose. Thus, further studies could prove their usefulness as hyperglycemia-independent biomarkers for T1D in early stages or non-fasting conditions.

Interestingly, we did not detect significant changes in the levels of reduced and oxidized glutathione (GSH and GSSG), although we detected a tendency of GSH levels to decrease, and the glutathione pathway seemed to be affected in enrichment analysis (Figure 4). Decreased GSH and increased GSSG levels have been previously detected in plasma [27,47,48] and RBCs [49,50] of T2D patients. This alteration was attributed to a decrease in the activities of the enzymes involved in GSH synthesis [49], a decrease in the transport rate of oxidized glutathione (GSSG) from RBCs [51], an enhanced sorbitol pathway [52], and the increase of oxidative stress produced by the elevated generation of ROS linked to T2D [53]. Our result was coherent with the fact that a decrease of GSH in blood of T1D patients was detected less frequently [47,54]. Thus, the glutathione pathway seems to be less affected by T1D.

We also did not detect significant differences in 2,3-Bisphosphoglyceric acid (2,3-BPG) levels, which have been described for T2D [25]. 2,3-BPG regulates oxygen absorption by hemoglobin, and its alteration can be directly associated to hypoxia and related to oxidative stress [55]. Thus, our study indicates lower hypoxia in organs and tissues related to RBCs and reactive oxygen species (ROS) levels in RBCs in T1D patients, in comparison with T2D. This result is in accordance with the lower impact on glutathione metabolism that we detected for T1D patients.

Regarding the lipidomic profile of RBCs, we found very significant differences between T1D patients and healthy controls. This result was in coherence with previous works that detected alterations in lipid metabolism associated with T1D and T2D [56,57,58]. Specifically, we detected a very clear decrease in lipid chain length in combination with an increase in lipid methyl groups, which could be an indication for lipid peroxidation, as was observed previously in peripheral blood mononuclear cells [59]. On the other hand, the increase in methyl lipid groups may also be related to the higher lipid accumulation of T1D patients due to the alterations of insulin release from pancreatic beta cells that inhibit the glucose uptake in cells [60]. In addition, we detected changes in the phospholipid phosphatidylcholine, as well as for the related metabolites phosphocholine and glycerophosphocholine (GPC), which are all important constituents of the cell membrane. As mentioned, diabetes negatively affects the membrane and the shape of RBCs [19,25,56], which is in coherence with these results. This alteration also was shown in RBCs of T2D patients [25].

Our study is only a first pilot study, and has several limitations that will have to be overcome in additional studies. For instance, although our study groups were sex-matched to compensate for metabolic differences between men and women, the low number of samples did not allow us to perform sex-specific studies to identify specific alterations for each sex group. Additionally, the effect of many other factors, such as age, comorbidities, or treatments, should be studied.

Nevertheless, we show robust differences in the metabolic and lipidomic profile of RBCs from T1D patients that provide a valuable starting point for the identification of biomarkers for an earlier diagnosis or monitoring of patients with T1D. Several of the detected metabolic changes are similar to alterations detected previously in the RBCs of T2D patients. Further studies are needed to confirm if these alterations are also related to the presence of IR in T1D patients.

## Figures and Tables

**Figure 1 jcm-12-00556-f001:**
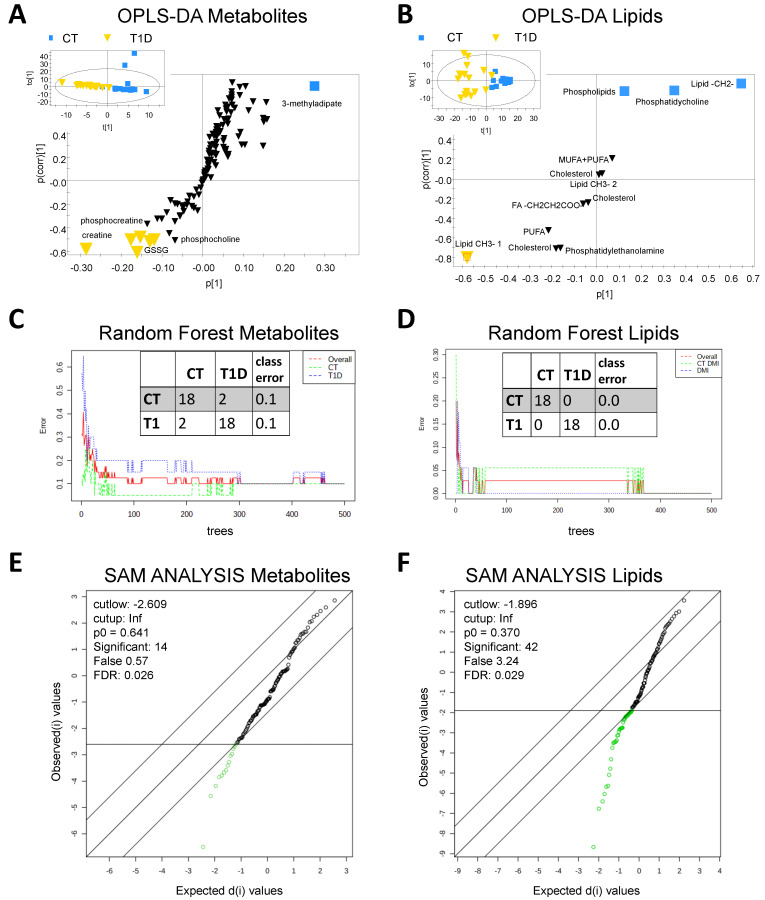
(**A**) Score and S-plot from OPLS-DA analysis comparison of the aqueous RBC profile of T1DM patients and healthy controls. RY(cum) = 0.77, Q(cum) = 0.64, CV-ANOVA = 3.9 × 10^−6^. (**B**) Score and S-plot from OPLS-DA analysis comparison of the lipidomic RBC profile of T1DM patients and healthy controls. RY(cum) = 0.83, Q(cum) = 0.66, CV-ANOVA = 1.1 × 10^−6^. (**C**) Random Forest classification of aqueous RBC metabolites of T1D patients and healthy controls, 500 trees, 7 predictors, randomness. (**D**) Random Forest classification of lipidic RBC metabolites of T1D patients and healthy controls, 500 trees, 5 predictors, fixed seed. (**E**) SAM analysis of aqueous metabolites, Delta = 1.4, FDR 0.026, significant compounds: 14. (**F**) SAM analysis of lipidic metabolites, Delta = 0.1, FDR 0.025, significant compounds: 10.

**Figure 2 jcm-12-00556-f002:**
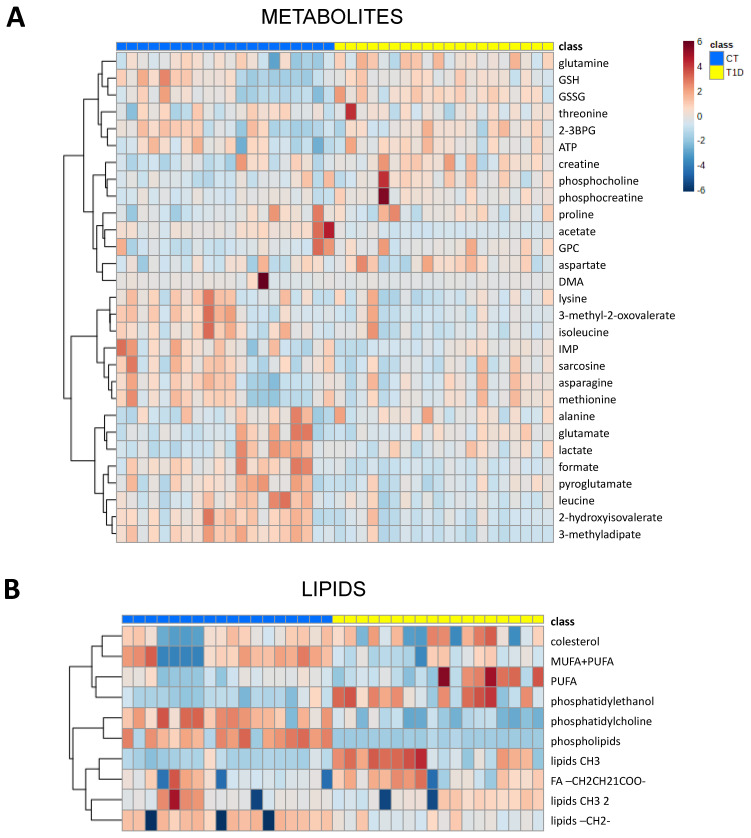
Heatmap analysis performed with the Ward clustering algorithm and ANOVA test. (**A**) Heatmap showing main aqueous metabolites. (**B**) Heatmap showing the main lipid groups. GSH = reduced glutathione, GSSG = oxidized glutathione, 2,3-BPG = 2,3-Bisphosphoglyceric acid, GPC = glycerophosphocholine, DMA = dimethylamine, IMP = inosinic acid, MUFA = monounsaturated fatty acids, PUFA = polyunsaturated fatty acids, CH3 = methyl group, FA = fatty acids, CH2 = lipid chain. The color scale refers to normalized concentration values.

**Figure 3 jcm-12-00556-f003:**
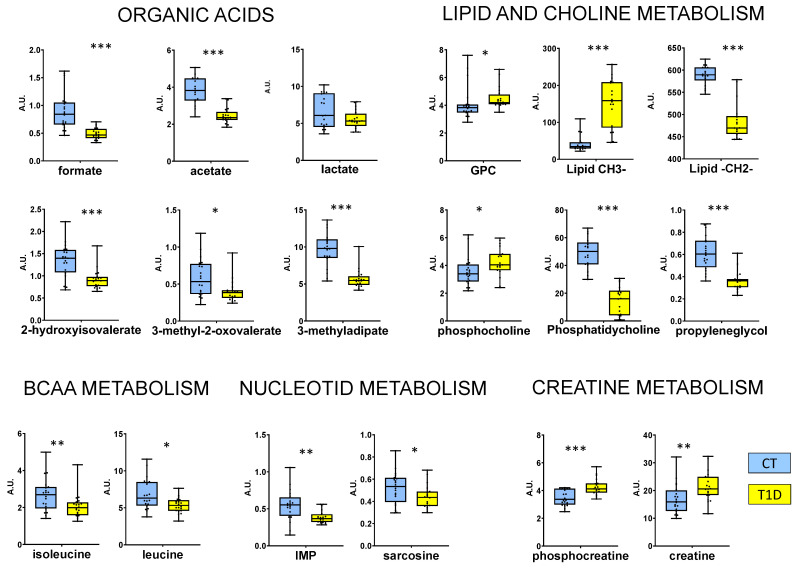
Box plots of RBC metabolites that showed relevant changes in T1D patients. GPC = glycerophosphocholine, CH3 = methyl group, CH2 = lipid chain, IMP = inosinic acid. Concentrations were of normalized total intensity. * = *p* < 0.05, ** = *p* < 0.01, *** = *p* < 0.001 in Wilcoxon test. A.U. = arbitrary units from normalized concentration values.

**Figure 4 jcm-12-00556-f004:**
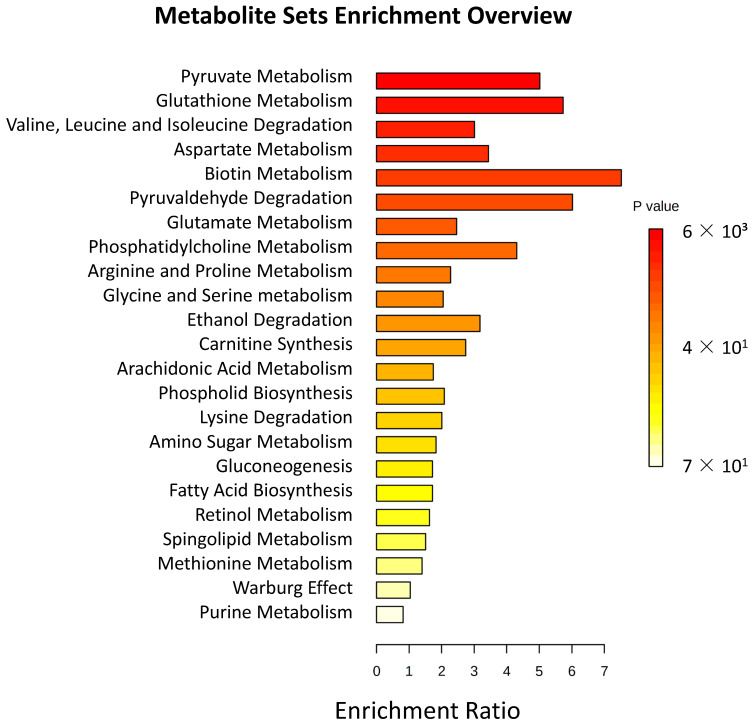
Pathway enrichment overview of metabolomic changes in RBCs of T1D obtained with Metaboanalyst.

**Table 1 jcm-12-00556-t001:** Main characteristics of the cohort of T1D patients.

	T1D	CT	*p*
Age (years)	32.3 ± 1.6	33.0 ± 2.4	0.821
Gender F/M	8/12	8/12	-
HbA1c (DCCT%)	8.00 ± 0.29	5.12 ± 0.06	<0.001
Glucose (mg/dL)	168.7 ± 18.3	82.4 ± 2.2	<0.001
BMI (kg/m^2^)	24.1 ± 1.0	23.4 ± 0.8	0.604
Cholesterol (mg/dL)	187 ± 10	185 ± 7	0.892
HDL (mg/dL)	57 ± 3	60 ± 3	0.440
LDL (mg/dL)	116 ± 9	109 ± 6	0.525
Triacylglycerides (mg/dL)	75 ± 9	84 ± 9	0.483

CT = healthy control, F/M = female/male, HbA1c = glycosylated hemoglobin levels, BMI = body mass index. Data are shown as mean ± SD. *p*-values were calculated with the Student’s *t*-test.

## Data Availability

The datasets generated and/or analysed during the current study are available in the Zenodo repository, “https://doi.org/10.5281/zenodo.6597902 (accessed on 2 September 2022)”.

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
