# Peer review of "The Footprint of Type 1 Diabetes on Red Blood Cells: A Metabolomic and Lipidomic Study"

_jcm, 2023, doi:10.3390/jcm12020556_

Round 1

Reviewer 1 Report

The manuscript is well-designed and written. However, I have some minor concerns that need to be addressed.

1. The authors have used "arbitrary units (A.U.)" to report their results (Fig 3 and 4). It is recommended to clarify how such units are interpreted in the manuscript.

2. Did the authors use a standard compound for NMR measurements to justify the accuracy of their measurements?

Author Response

Answers to reviewer 1

The manuscript is well-designed and written. However, I have some minor concerns that need to be addressed.

We appreciate this positive feedback.

  1. The authors have used "arbitrary units (A.U.)" to report their results (Fig 3 and 4). It is recommended to clarify how such units are interpreted in the manuscript.

We have now added to the figure caption that arbitrary units (A.U.) refers to normalized concentrations values. Further, we have completed the Data analysis section of Materials and Method explaining why it is important to work with normalized and not absolute concentration values, saying “For metabolite quantification, spectra were normalized to total intensity. This allows us to work with normalized concentration values and avoid the detection of changes related exclusively to differences in total sample amount and experimental error during the ex-traction process.”

  1. Did the authors use a standard compound for NMR measurements to justify the accuracy of their measurements?

Previously to all our NMR measurements we performed a parameter optimization with a standard sucrose sample to confirm an optimal resolution and sensitivity of our measurements. Furthermore, we acquired spectra of a quality control sample consisting in a mixture of metabolites (lactate, creatine, citric acid, phenylalanine and glucose) in regular intervals to check reproducibility. We have now completed the “NMR experiments” section of Materials and Methods with this information

Reviewer 2 Report

Herance et al. aim to search for new biomarkers of type 1 diabetes in red blood cells (RBCs). For this purpose, the author assessed the metabolomic and lipidomic profile of RBCs in patients with T1D by using NMR spectroscopy. T1D patients exhibited a different metabolic and lipidomic profile, specifically, T1D patients have lower levels of unsaturated fatty acids, phosphatidylcholines, and phospholipids as compared to the control subjects. Overall, this is an interesting topic and has an important clinical application for predicting the outcome of T1D.

Major

1. Criteria of type 1 diabetes identification/diagnosis. Usually, islet autoantibodies are used as a diagnostic criterion. If this is the case, do the T1D patients that were recruited in this study have the same islet autoantibody? If not, is there any metabolomic and lipidomic difference between the patients with different autoantibodies?

2. While the number of T1D patients or control subjects is 20, it would be necessary to look at the correlation between glucose levels and metabolites. For example, whether a higher glucose level is related to a higher creatine/ phosphocreatine level?

3. The author also tried to link the metabolic and lipidomic profiles of T1D with insulin resistance. However, there are no insulin levels or HOMA-IR in the study.

Minor

1. Does glucose level represent fasting blood glucose level or non-fasting blood glucose level?

2. Format: for the in-text citation, the citation should be placed before the final punctuation mark in a sentence. Please check and make changes.

For example,

Due to the latent onset of this disease, there is also a long pre-detection period, and up to one-half of cases in the population may be undiagnosed.(Soriguer et al. 2012)

T1D accounts for about 5%–10% of all cases of diabetes and is due to the autoimmune destruction of beta cells of the pancreas, leading to a total incapacity of producing insulin.(Mobasseri et al. 2020; Patterson et al. 2019)

3. Please add a citation

P2

In T1D, immune cells such as macrophages, which are the source of pro-inflammatory cytokines, migrate to pancreatic islet cells.

4. Consistency: for example, beta cell or b cell

Author Response

Answers to reviewer 2

Herance et al. aim to search for new biomarkers of type 1 diabetes in red blood cells (RBCs). For this purpose, the author assessed the metabolomic and lipidomic profile of RBCs in patients with T1D by using NMR spectroscopy. T1D patients exhibited a different metabolic and lipidomic profile, specifically, T1D patients have lower levels of unsaturated fatty acids, phosphatidylcholines, and phospholipids as compared to the control subjects. Overall, this is an interesting topic and has an important clinical application for predicting the outcome of T1D

We appreciate this positive feedback.

Major

  1. Criteria of type 1 diabetes identification/diagnosis. Usually, islet autoantibodies are used as a diagnostic criterion. If this is the case, do the T1D patients that were recruited in this study have the same islet autoantibody? If not, is there any metabolomic and lipidomic difference between the patients with different autoantibodies?

In our study, all the patients had positive islet autoantibodies; anti GAD-65 (>50 IU/ml) was positive in all cases. To clarify this, a sentence has been added to the manuscript in the “Study design and subjects” section of Materials and Methods.

  1. While the number of T1D patients or control subjects is 20, it would be necessary to look at the correlation between glucose levels and metabolites. For example, whether a higher glucose level is related to a higher creatine/ phosphocreatine level?

We thank the reviewer for this interesting suggestion. We have now included a PLS regression analysis of the lipidomic and metabolomic profile versus glucose levels, to see which of the altered metabolites have a direct correlation with blood glucose levels. We have added these new data in the results section (P8) and the discussion section (P10), and have created an additional supplementary material to include additional figures and tables.

  1. The author also tried to link the metabolic and lipidomic profiles of T1D with insulin resistance. However, there are no insulin levels or HOMA-IR in the study.

In the case of T1D, HOMA-IR is not recommended as a surrogate of insulin resistance, since the insulin that is detected in blood is not endogenous. Our statements related insulin resistance are based on several publications of the literature that prove that insulin resistance is also present in patients with T1D. This point is discussed more in detail in the third and fourth paragraph of the introduction that we have completed with additional references. We have also changed the wording of the last paragraph of the discussion, clarifying the further studies are required to confirm that the metabolic changes are related to the presence of IR.

  1. Does glucose level represent fasting blood glucose level or non-fasting blood glucose level?

The glucose level represents fasting blood glucose. We mention this in the section “Peripheral blood samples” of Materials and Methods: “Briefly, peripheral blood was collected under fasting conditions, stored at 4° C and processed within the first hour.” and now we have also included a comment in the “Study design and subjects”, for more clarity.

  1. Format: for the in-text citation, the citation should be placed before the final punctuation mark in a sentence. Please check and make changes.

For example,

Due to the latent onset of this disease, there is also a long pre-detection period, and up to one-half of cases in the population may be undiagnosed.(Soriguer et al. 2012)

T1D accounts for about 5%–10% of all cases of diabetes and is due to the autoimmune destruction of beta cells of the pancreas, leading to a total incapacity of producing insulin.(Mobasseri et al. 2020; Patterson et al. 2019)

We thank the reviewer for correction and have changed now the order of citations and final punctuation marks, when necessary.

  1. Please add a citation

P2

In T1D, immune cells such as macrophages, which are the source of pro-inflammatory cytokines, migrate to pancreatic islet cells.

We have now added the following citation after the sentence: “Citro, Antonio, Francesco Campo, Erica Dugnani, and Lorenzo Piemonti. 2021. “Innate Immunity Mediated Inflammation and Beta Cell Function: Neighbors or Enemies?” Frontiers in Endocrinology 11(February): 1–7.”

  1. Consistency: for example, beta cell or b-cell

We thank the reviewer for correction and have checked the consistency of wording in our article.

Round 2

Reviewer 2 Report

I have no further questions.